# Zebrafish Model in Illuminating the Complexities of Post-Traumatic Stress Disorders: A Unique Research Tool

**DOI:** 10.3390/ijms25094895

**Published:** 2024-04-30

**Authors:** Raed M. Al-Zoubi, Haya Abu-Hijleh, Ahmad Zarour, Zain Z. Zakaria, Aksam Yassin, Abdulla A. Al-Ansari, Maha Al-Asmakh, Hiba Bawadi

**Affiliations:** 1Department of Chemistry, Jordan University of Science and Technology, P.O. Box 3030, Irbid 22110, Jordan; rmzoubi@just.edu.jo; 2Surgical Research Section, Department of Surgery, Hamad Medical Corporation, Doha P.O. Box 3050, Qatar; yaksam@hamad.qa (A.Y.); aalansari1@hamad.qa (A.A.A.-A.); 3Department of Biomedical Sciences, QU-Health, College of Health Sciences, Qatar University, Doha 2713, Qatar; 4Department of Human Nutrition, QU-Health, College of Health Sciences, Qatar University, Doha 2713, Qatar; ha1601254@qu.edu.qa (H.A.-H.); maha.alasmakh@qu.edu.qa (M.A.-A.); 5Department of Surgery, Acute Care Surgery, Hamad Medical Corporation, Doha P.O. Box 3050, Qatar; azarour@hamad.qa; 6Vice President for Medical and Health Sciences Office, QU-Health, Qatar University, Doha 2713, Qatar; zain.zakaria@qu.edu.qa; 7Center of Medicine and Health Sciences, Dresden International University, 01069 Dresden, Germany; 8Biomedical Research Center, Qatar University, Doha P.O. Box 2713, Qatar

**Keywords:** associative learning, post-traumatic stress disorder, stress, time-dependent sensitization

## Abstract

Post-traumatic stress disorder (PTSD) is a debilitating psychological condition that may develop in certain individuals following exposure to life-threatening or traumatic events. Distressing symptoms, including flashbacks, are characterized by disrupted stress responses, fear, anxiety, avoidance tendencies, and disturbances in sleep patterns. The enduring effects of PTSD can profoundly impact personal and familial relationships, as well as social, medical, and financial stability. The prevalence of PTSD varies among different populations and is influenced by the nature of the traumatic event. Recently, zebrafish have emerged as a valuable model organism in studying various conditions and disorders. Zebrafish display robust behavioral patterns that can be effectively quantified using advanced video-tracking tools. Due to their relatively simple nervous system compared to humans, zebrafish are particularly well suited for behavioral investigations. These unique characteristics make zebrafish an appealing model for exploring the underlying molecular and genetic mechanisms that govern behavior, thus offering a powerful comparative platform for gaining deeper insights into PTSD. This review article aims to provide updates on the pathophysiology of PTSD and the genetic responses associated with psychological stress. Additionally, it highlights the significance of zebrafish behavior as a valuable tool for comprehending PTSD better. By leveraging zebrafish as a model organism, researchers can potentially uncover novel therapeutic interventions for the treatment of PTSD and contribute to a more comprehensive understanding of this complex condition.

## 1. Introduction

Post-traumatic stress disorder (PTSD) is a severe mental condition that can develop in individuals after a personal encounter with a life-threatening or traumatic event, such as a natural disaster, serious accident, physical assault, or another form of violence [1]. Flashbacks, dysregulated stress, fear and anxiety, avoidance of reminders of the traumatic event, and insomnia are all symptoms of PTSD [1]. Addressing and overcoming PTSD is crucial for preventing long-term detrimental effects on personal and familial relationships, as well as burdens on social, medical, and financial stability [2]. The prevalence of PTSD varies depending on the population and the nature of the traumatic event [3]. According to the National Center for PTSD, an estimated 6–8% of the US population may experience PTSD at some juncture in their lives [3]. The prevalence is higher among individuals who have experienced traumatic events such as military combat, sexual assault, and physical assault [3].

In recent years, zebrafish have emerged as a prominent model organism in scientific research, particularly for studying various conditions and disorders. Their observable behavioral patterns have become increasingly quantifiable with advancements in video-tracking technology. While zebrafish possess a comparatively simpler nervous system than humans, their genetic makeup is well characterized, sharing a nucleotide sequence akin to most vertebrates [4]. Notably, the zebrafish genome has been fully sequenced, revealing an 87% similarity to the human genome [4,5], with approximately 70% of their genome being orthologous to humans [4,6]. This genetic resemblance suggests the presence of numerous genes associated with human diseases in zebrafish, making them valuable for investigating mutations observed in human subjects. Consequently, zebrafish serve as an effective model for exploring the molecular and genetic mechanisms underlying behavior and offer a powerful comparative tool for advancing our understanding of conditions such as PTSD [7,8].

## 2. Zebrafish as an Emerging Model Organism in PTSD Research

Zebrafish have become valuable models in studying stress-related disorders like post-traumatic stress disorder (PTSD) due to their genetic tractability, rapid reproduction, and shared physiological and molecular pathways with humans [9,10]. While zebrafish do not experience PTSD in the same way humans do, they exhibit stress responses that make them suitable for studying the underlying mechanisms and potential treatments [11]. Zebrafish can exhibit behavioral and molecular alterations in response to various stressors [12]. Physical stressors like temperature fluctuations and water quality changes can impact swimming patterns and gene expression [13]. Chemical stressors, including pollutants and toxins, disrupt physiological processes and trigger molecular stress responses [14]. Social stressors, such as aggression and social interactions, affect behavior and neural pathways related to social behaviors [15]. Psychological stressors like isolation and novel environments activate the stress response system, influencing behavior and gene expression [16]. Infections and pathogens induce stress and alter immune responses, leading to molecular changes [17]. Nutritional stress affects growth, reproduction, and metabolism, resulting in behavioral and molecular adaptations [12,13]. Long-term exposure to environmental changes can lead to chronic stress, causing persistent behavioral and molecular adjustments [18,19]. These responses are often adaptive and aimed at helping the fish cope with the specific stressor and maintain homeostasis [12]. Researchers often study these responses in zebrafish to gain insights into stress biology and potential human health implications.

When exposed to stressors, zebrafish exhibit hormonal changes, altered behaviors, and neural circuitry responses that mimic aspects of human stress responses [13]. Their transparent embryos and larvae are particularly useful for studying early-life stress and neural development [20]. Additionally, zebrafish models offer ethical advantages, as they allow for genetic and pharmacological manipulations with fewer ethical concerns than mammalian models [21]. While zebrafish may not fully replicate the cognitive and emotional aspects of PTSD, they provide valuable insights into the underlying mechanisms and potential treatments, complementing research conducted in other species and humans [22].

Despite these advantages, it is important to note that zebrafish models have limitations such as anatomical disparities in the central nervous system (CNS), a restricted availability of research tools such as antibodies for testing, and the existence of duplicate genes that may pose challenges in terms of modeling or silencing. Moreover, they lack the complex cognitive and emotional aspects of PTSD that humans experience [22]. However, they provide a valuable platform for studying the molecular and physiological underpinnings of stress responses and can complement research conducted in other animal models and humans. Table 1 highlights the commonalities between zebrafish and humans in PTSD research.

It is essential to highlight that zebrafish models for PTSD research are still a relatively new area of study, and ongoing research is essential to validate and expand the understanding of their relevance in this field.

In summary, zebrafish models are scientifically reliable tools for investigating the physiological and genetic aspects of stress responses related to PTSD. While they may not fully replicate the human condition, their advantages make them valuable contributors to our understanding of the mechanisms underlying stress-related disorders and the development of potential treatments.

## 3. Pathophysiology of PTSD

The human body responds to stress with a short-term stimulation of the hypothalamic–pituitary–adrenal (HPA) axis and the release of stress hormones and other mediators, such as neurotransmitters and cytokines, which play an essential role in adaptation to and protection against stressors [1]. The HPA axis regulates the body’s cortisol, a glucocorticoid, and has several effects throughout the body that help to support the stress response, including inhibiting insulin and increasing glucose availability, regulating immune system functions, and influencing electrolyte balance [31]. During stressful events, norepinephrine and inputs from the hippocampus, medial prefrontal cortex, and amygdala act on neurons in the hypothalamic paraventricular nucleus (PVN) that contain corticotropin-releasing hormone (CRH) [31]. The HPA axis is thus activated by the release of CRH, which travels to the anterior pituitary, where it stimulates the release of adrenocorticotropin (ACTH) into the systemic circulation. The ACTH stimulates the release of cortisol, the primary HPA axis effector molecule [31]. The crucial aspect of the HPA axis is the negative feedback that cortisol generates. The cortisol activity in healthy subjects has a short-term impact, as it travels to the pituitary gland and hypothalamus to reduce CRH and ACTH release [32]. As a result, cortisol is both the primary molecule that allows the stress response and the primary inhibitor of ongoing HPA axis activity [32].

The organism’s ability to maintain stability through change and continuous adjustments of the internal physiological factors for healthy functioning is called allostasis [33]. On the contrary, allostatic load is the maladaptive response to stressors, resulting in stress hormone dysregulation and a failure to terminate HPA axis activation [33]. Allostatic load causes physical and mental health consequences, such as inducing structural and functional brain changes in individuals, which may predispose them to developing neuropsychiatric disorders such as PTSD [33]. Additionally, the adrenal medulla instantly releases noradrenaline and adrenaline as part of the sympathetic nervous system (SNS) response following exposure to stress [34]. The release of cortisol that follows initially intensifies the SNS response but restrains it through negative feedback inhibition [34]. These processes are crucial for typical acute stress response and subsequent recovery. However, in individuals with PTSD, these processes are disturbed, causing the stress response to persist indefinitely and ultimately leading to a chronic stress response contributing to allostatic load [34]. Furthermore, this can lead to prolonged activation of the glucocorticoid receptor, which may result in the atrophy of the hippocampus caused by the release of nitric oxide (NO) through glutamate-mediated pathways [35]. Figure 1 summarizes the physiological mechanisms of the stress response and allostatic load and their implications for PTSD.

The he hypothalamo–pituitary–interrenal (HPI) axis in zebrafish has a similar role in stress response to that of the HPA axis in humans [36]. The zebrafish HPI and human HPA axes are stimulated to release cortisol when exposed to a stressor. In humans, glucocorticoids play an important role in retrieving traumatic memories. Studies on human subjects reported increased glucocorticoids, such as cortisol, to improve memory consolidation while disrupting memory retrieval [37]. In line with that, an experimental study on the zebrafish model showed that zebrafish exposed to stressors also have a stimulated release of cortisol. The increased glucocorticoid release also negatively affects memory consolidation and retrieval, similar to what occurs in humans [38].

Moreover, an experimental study assessed the effect of acute psychological stress on spatial and cued memory in adult zebrafish using the alarm pheromone or Indian leaf fish exposure. The results of this study showed that in the zebrafish plus-maze test, acute single inescapable stress significantly impaired spatial and cued memory, reducing correct arm entries and time spent in the target arm [39], strongly supporting the use of zebrafish in research related to neurobehavioral disorders. Therefore, the similarities between human and zebrafish responses to stressors, as indicated by evidence from the literature, support the use of zebrafish as a model for stress- and anxiety-related behavior.

**Figure 1 ijms-25-04895-f001:**
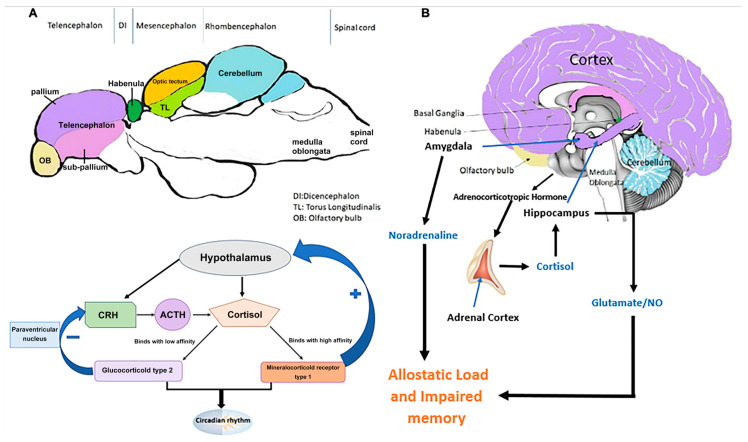
Comparison of (**A**) mid-sagittal sections of the zebrafish and (**B**) human brains, highlighting regions crucial for learning and memory. The figure visualizes the neuroendocrine and physiological pathways mediating the stress response and allostatic load, exploring their potential links to PTSD development in both species, modified by [40]. Hypothal: hypothalamus; PG: pituitary gland, GR: glucocorticoid receptors; ACTH: adrenocorticotropic hormone; AC: adrenal cortex; HPC: hippocampus; NO: nitric oxide.

## 4. Neuroanatomical and Neurochemical Insights from Zebrafish PTSD Studies

Zebrafish and humans exhibit shared brain structures, including the amygdala and hippocampus, which play a pivotal role in fear processing and stress response. Studies on zebrafish have identified modifications in neural circuits within these brain regions following exposure to stress or trauma [41]. Neurotransmitter systems have also been examined, with dysregulation of serotonin 5-hydroxytryptophan (5-HT) linked to anxiety-like behavior in zebrafish subjected to stress paradigms [42]. Moreover, investigations into the HPA axis have revealed insights into dysregulation after stress exposure, leading to altered cortisol levels and stress-related behaviors [43,44].

Zebrafish emerge as a promising model organism for unraveling the intricacies of PTSD, as has been suggested by other published scientific papers [5,39,45,46]. A study delves into the impact of early prednisolone exposure on zebrafish larvae, revealing notable changes in CpGI2 methylation within the nr3c1 promoter region [47]. These methylation alterations influence GR expression, ultimately leading to abnormal behavioral patterns in adult male zebrafish [47]. This highlights a significant surge in cortisol levels among adult male zebrafish subjected to early prednisolone exposure. Intriguingly, there is a simultaneous decrease in nr3c1 gene expression, a key player in the negative feedback regulation of the HPA axis. This compromised regulatory function in adult male zebrafish mirrors observations in individuals with psychiatric disorders, where cortisol levels and CRH activity are elevated [48]. Drawing parallels with findings in human studies, the research underscores the impact of early adverse life experiences on nr3c1 gene methylation, heightening the susceptibility to PTSD in a manner proportional to the severity of such early traumas [49,50]. Analyzing adult male zebrafish brains, the study explores genes associated with the HPA axis and neurotransmitter secretion. This comprehensive approach enhances our understanding of the abnormal behavior in zebrafish induced by central nervous system disorders, thereby establishing a robust psychopathological model [47].

Zebrafish studies have further explored neuroinflammation and oxidative stress, indicating potential contributions to neuronal damage and cognitive impairments associated with PTSD-like behaviors [51]. Epigenetic modifications, such as DNA methylation and histone modifications, have also been examined, as they may underlie the lasting effects of stress on gene expression and behavior [11,47,52]. These neurobiological findings contribute significantly to our understanding of stress-related disorders, and while the field continues to evolve, further research on zebrafish PTSD models is needed to elucidate the complex mechanisms involved.

## 5. Post-Traumatic Stress Disorder (PTSD) in Zebrafish: Learning Paradigms and Behavioral Aspects

Researchers have developed several behavioral paradigms to assess PTSD-like behavioral aspects in zebrafish, aiming to model aspects of human PTSD symptoms such as anxiety-like behaviors, fear responses, and avoidance. Commonly used paradigms include the light–dark box test, open field test, novel tank diving test, shoaling test, startle response, and learned helplessness paradigm [23,30,53]. These tests evaluate anxiety levels, locomotor activity, thigmotaxis, social interactions, startle responses, and learned helplessness in zebrafish. While providing valuable insights, no single test fully replicates human PTSD; therefore, researchers often employ a combination of tests to better comprehend the complex behavioral aspects associated with PTSD in zebrafish [23,30,53].

Learning paradigms are broadly classified into associative and non-associative learning [54,55]. Associative learning involves the establishment of connections between two or more stimuli, requiring an active learning process where the animal learns to associate events or stimuli, leading to behavior changes based on this acquired knowledge [54]. This type of learning involves multiple stimuli, often including a neutral and a reinforcing or punishing stimulus, and is characterized by the formation of long-lasting memories that facilitate future behavioral adaptations. Examples of associative learning include classical conditioning (aversive conditioning—fear), operant conditioning (positive reward-based conditioning), and social learning [55,56]. Non-associative learning, in contrast, focuses on the animal’s response to a single stimulus or event without the necessity of forming explicit associations. This form of learning is more passive, with behavior modifications occurring automatically due to repeated exposure or changes in stimulus intensity [54]. Typically, the effects of non-associative learning are short-term and specific to the encountered stimulus, including habituation, sensitization (discrimination or novelty), perceptual learning, priming, and recognition memory [21,55].

### 5.1. Fear and Anxiety

Fear or anxiety is a common behavioral condition linked to a traumatic experience [57]. Stress responses stimulate a complex set of physiologic reactions and are considered a significant factor that can contribute to anxiety and psychiatric disorders [57].

Zebrafish are increasingly recognized as a promising model organism for anxiety and stress research due to their dependable and easily measurable cortisol stress response [58,59,60]. Zebrafish can experience acute stress upon exposure to air, changing water PH, salinity, temperature, crowding, handling, or exposure to bright light [13]. Other methods of inducing anxiety in zebrafish include alarm substances from conspecifics and net chasing [13]. In response to these stressors, the zebrafish might exhibit anxiety behaviors such as erratic locomotion (increased distance and speed), freezing, avoidance of bright areas/light exposure, memory deficit, and reduced cognition [13]. As previously mentioned, the zebrafish HPI axis is stimulated under these conditions to release cortisol, just like what occurs in humans, which makes the zebrafish an excellent model for studying neuroscience and psychological conditions [13,61].

The novel tank diving task (NTT) and the light–dark test (LDT) are the two most widely used assays for investigating anxiety-like behavior in zebrafish. The novel open tank task is designed to examine behavior in response to stress. In this task, zebrafish may exhibit increased activity that becomes habitual over time, as well as fear-related behaviors. In zebrafish, as in many other species, novelty induces strong anxiety responses [53]. The novel tank diving test uses zebrafish instincts to seek protection in a strange environment by diving deep and staying at the bottom until they feel safe and secure enough to explore [62]. Researchers can use this model to investigate and compare behavioral parameters to assess anxiety. Such parameters include latency to swim to the upper part of the tank, transitions into the upper half of the tank, erratic movements, and freezing bouts [63]. A longer latency manifests anxiety about swimming to the upper part, less time spent at the top, and increased erratic movements and freezing [53,63].

In comparison, the dark–light tank test measures the fish’s preference for the tank’s light or dark areas. Usually, the fish exhibits an anxiety-related behavior termed scototaxis. Scototaxis is the term used to describe zebrafish’s preference for dark environments over brightly lit ones. This trait has been proposed to represent a species-specific defense strategy for avoiding predators. However, a third test also used to detect anxiety in zebrafish is the dark–light transition test, a different essay that measures the change in locomotion resulting from sudden illumination.

Blaster et al. used the dark–light tank approach to analyze the defensive behavior of animals in a laboratory test situation [64]. In this strategy, the researchers observe the animal’s behavior in the test situation and analyze its ethology (the study of its typical behavior and response in its natural environment) to identify patterns and dimensions of defensive behavior that are relevant from an ethological perspective. By identifying these ethologically relevant dimensions of defensive behavior, researchers can better understand the animal’s natural behavior and its response to different stimuli in the laboratory setting [64].

In this previous study, it was found that zebrafish preferred a dark environment. They conducted an experiment that confined the animals to each compartment and observed their behavior. After the confinement, they conducted another preference test. The animals were divided into “high-avoidant” and “low-avoidant” based on how much time they spent in the white compartment, using a median split of their pre- and post-test scores. When confined to the white compartment, high-avoidance animals froze in place, but low-avoidance animals did not [64]. There was a weak relationship between erratic movement and avoidance, and thigmotaxis and locomotor behavior were not good predictors of high or low avoidance. As a result, the researchers concluded that freezing is an excellent way to measure anxiety caused by being in the white compartment, while erratic movement, thigmotaxis, and locomotor behavior are not good predictors of avoidance levels [64].

Moreover, zebrafish have been used as a model for anxiety in relation to different exposures [60,65,66]. Baicalein, a plant flavonoid, contributes to many biological processes, including neuroprotection [67]. Zebrafish were employed in an experimental investigation to assess the chemical exposure of Baicalein as an anti-anxiety substance under the influence of acute and unexpected chronic stress. Then, LDT and NTT were used to evaluate the effect. The results of this study revealed that Baicalein poses a therapeutic advantage in reversing the adverse effects of both acute and unexpected chronic stress, making it a promising therapeutic approach for stress [60].

Fish behavior, including reproductive, defensive, social, and migratory behaviors, is significantly regulated by odor [23]. Anosmia, a lack of olfaction, can be experimentally induced in fish, thus limiting their capacity to react to different olfactory cues [23]. In a previous study, adult zebrafish were used to study the effects of experimental lidocaine-induced anosmia on anxiety-like behavior and whole-body cortisol levels [65]. The findings demonstrate that experimentally induced anosmia diminishes fluoxetine’s anxiolytic-like behavioral effects and appears to interact with stress’s anxiogenic effects, which are also shown to match cortisol responses in zebrafish [65]. These results present scientific evidence that transitory anosmia affects the physiology and behavior of anxiety in adult zebrafish.

A recent study developed a chronic unpredictable stress protocol appropriate for early life stages in zebrafish and investigated how chronic stress exposure affects zebrafish development and anxiety-related behaviors [16,51]. In this study, zebrafish were exposed to a series of irregular and unpredictable mild stressors. The stressed fish showed decreased exploration of a novel environment one day after stress and increased responsiveness to dark–light transition two days later, indicating increased anxiety-related behaviors [16]. The decrease in exploration behavior induced by stress lasted at least three days before returning to control levels one week later. Furthermore, stressed fish were 8% smaller than control siblings two days after stress and returned to control levels within one week. Overall, the findings of this study show that young zebrafish exposed to chronic unpredictable stress exhibit growth and behavioral changes similar to those reported in rodent models [16].

Another experimental study used zebrafish to study the association between a particular gene and behavioral outcomes. Researchers investigated the relationship between the Neuropeptide Y (NPY) gene and emotional behaviors [68]. The pleiotropic gene NPY is associated with high levels of conscientiousness and is involved in stress resilience [69]. In this experiment, the NPY gene in zebrafish was knocked down, and the NPY-deficient zebrafish displayed several anxiety-like behaviors, including decreased social interaction in the mirror test and decreased locomotion in the black–white test. In the mirror test, the acute cold stress-treated NPY-deficient zebrafish displayed anxiety-like behaviors such as staying stationary and swimming along the side of the tank [68]. Furthermore, NPY-deficient zebrafish had significantly higher levels of expression of anxiety-related genes (orx and cck) and catecholamine production than wild-type fish [68].

### 5.2. Associative Learning

Learning and memory are characteristics shared by all animals, allowing the organism to respond to changing environments in a plastic manner. They contribute to adaptation in a fraction of an organism’s lifetime, rather than waiting thousands of generations for genetic evolution and adaptation to environmental changes [70]. Learning is the understanding of experience-dependent changes in behavior, whereas memory is the consolidation, storage, and recall of acquired information. They form a temporal series of neurobiological mechanisms that result in experience-dependent changes in behavior [70].

Associative learning is one of the most extensively researched and complex types of learning and memory. Acquiring a temporal and/or causal relationship between at least two stimuli is required for associative learning. When an organism learns the association between two stimuli, the unconditioned stimulus and the conditioned stimulus [71], this is an example of simple associative learning. Zebrafish have the ability to learn and adapt their behavior in response to specific environmental cues. Zebrafish are often used in research as a model organism for studying basic principles of learning and memory, as they learn to associate a specific stimulus with a particular outcome or reward. For example, a zebrafish may learn to associate the presence of food with a particular visual cue or odor and subsequently respond to that cue by approaching the food [72].

The other type is “relational learning”, which is one of the most complex forms of associative learning in which the animal learns loose relationships between a potentially large number of cues or stimuli [73]. Episodic memory and spatial learning are two examples of the latter [73].

In general, episodic memory is defined as the retrieval of a memory trace of a past experience that can be identified based on what happened, the context in which the event occurred, and when the event occurred in subjective time [74]. An experimental study was conducted to test episodic memory in zebrafish [75]. The results of this study show that when zebrafish are presented with a familiar object in a familiar context but in a different quadrant within that context, fish spend more time in that quadrant. These findings indicate that the fish can recall what object they previously saw and where and when it was presented [75]. Moreover, spatial learning occurs when a zebrafish learns to navigate and remember the spatial layout of its environment [56]. Zebrafish have been shown to use visual landmarks and spatial cues to navigate their environment and can remember the location of food or other rewards in a complex maze [56].

The learning abilities of zebrafish are thought to be related to the organization and function of their brains. The zebrafish brain is relatively simple compared to the brains of mammals, but it contains many of the same basic structures and functions, including regions involved in learning and memory. For example, the telencephalon, the largest region of the zebrafish brain, is thought to be involved in spatial learning and other forms of cognitive processing [76].

Zebrafish offer significant advantages as a model organism for investigating a wide range of learning behaviors and the underlying neural mechanisms. Their relatively simple nervous system, easy maintenance in laboratory settings, and genetic traceability make them valuable resources for researchers studying learning and memory. The behavior of zebrafish in response to learning tasks allows for examining neural mechanisms related to learning and memory and developing potential treatments for neurological disorders that affect human learning and memory.

### 5.3. Time-Dependent Sensitization (TDS)

Time-dependent sensitization (TDS) is a phenomenon observed in zebrafish that refers to the gradual enhancement of the behavioral response to repeated exposure to a stimulus over time [77]. This means that with each exposure to a stimulus, the response of the zebrafish becomes more robust and more persistent. TDS has been observed in various behavioral responses in zebrafish, including swimming activity, shoaling behavior, and aggression [78]. For example, if zebrafish are repeatedly exposed to a predator odor, they may initially show a weak response, but with each subsequent exposure, their response becomes more potent and more persistent, even in the absence of the predator odor.

TDS is thought to be mediated by changes in the nervous system, specifically changes in the sensitivity of neural circuits that respond to the stimulus [78]. These changes may involve alterations in synaptic plasticity, neurotransmitter release, and receptor expression. TDS has been studied in zebrafish as a model system for understanding the neural mechanisms underlying learning and memory and the processes that contribute to the development of addiction and other behavioral disorders [78]. It is also being explored as a potential tool for assessing the effects of environmental toxins and other stressors on the behavior and health of aquatic animals.

TDS, as a stress–restress animal model of PTSD, is well validated, and it involves inducing an initial life-threatening trauma and then subjecting the animal to repeated exposure to the original stress and is known to produce long-lasting changes in the central nervous system (CNS). Over time, this may result in alterations in locomotor activity [79,80,81] and behavior that is associated with PTSD, including an intensified startle response [9] and cognitive impairments [82]. Restress is triggered by having flashbacks of the traumatic event, and each time the organism relives it, it releases cortisol. As the HPI axis fails to adapt, adrenal insufficiency is often observed in conjunction with this phenomenon [66,83].

In the TDS model context, exposure to stress triggers the sensitization of neurobiological mechanisms and a rise in negative feedback inhibition. Consequently, this leads to heightened fear response and cognitive impairments, which become evident after consolidation [82,84]. TDS can lead to the inhibition of the HPA axis and subsequent hypocortisolism [82]. The disturbed neuroendocrine stress response observed in individuals with PTSD and contextual cues have been shown to cause short-term alterations in the HPA axis and regional brain monoamines. These changes can worsen behavioral symptoms such as anxiety and memory deficits over time and ultimately contribute to long-term dysfunction [9,79,84]. The TDS model closely resembles the lived experience and clinical presentation of PTSD due to the specificity of the neurobiological phenotypes and the time-dependent nature of the alterations it induces [84,85]. Figure 2 summarizes zebrafish learning paradigms.

## 6. Genetic and Molecular Mechanisms Underlying PTSD-like Responses in Zebrafish

MicroRNAs (miRNAs) have emerged as essential regulators of gene expression in the context of stress-related behaviors in zebrafish [86]. Several studies have demonstrated the involvement of specific miRNAs, such as miR-133b, in modulating signaling pathways associated with stress and inflammation [86]. These findings underscore the complexity of zebrafish’s genetic and molecular networks underlying stress responses and provide valuable insights into the neurobiological basis of PTSD-like behaviors in this model organism.

Exploring genetic and molecular mechanisms in zebrafish has shed light on potential targets for therapeutic interventions and drug screening. Understanding the biological pathways involved in stress-related responses can aid in the development of novel treatments for PTSD and related disorders. However, it is important to acknowledge that the field of zebrafish PTSD research is still evolving, and further investigations are needed to fully comprehend the intricate interplay of these mechanisms and their relevance to human PTSD.

A specific protocol involving three common husbandry stressors, namely reducing the water level in the holding tank, using a net for a 5 min chase, and exposing the fish to air for 1 min, has been examined across various fish species, including zebrafish [87,88]. This protocol has been demonstrated to be a dependable method for eliciting molecular, endocrine, and physiological responses in specimens subjected to acute stressors [59]. In a study investigating the cortisol stress response dynamics and the underlying molecular regulation in adult zebrafish facing acute and prolonged stressors of varying nature, duration, and intensity, it was observed that zebrafish exhibited a swift and sustained elevation in cortisol levels, initiating around 15 min and returning to baseline levels approximately 2 h post-stress [59]. This rapid peak differs from findings in other fish species, where cortisol concentrations typically peak between 30 min and 4 h after stress [87,89,90]. The accelerated peak in zebrafish may be attributed to their rearing in higher water temperatures compared to cold freshwater or temperate marine fish species [91].

When neurons in the hypothalamus are triggered by environmental stressors, they respond by secreting corticotropin-releasing factor (CRF), which increases the production of cortisol [92]. Fish subjected to acute or chronic stressors exhibit changes in mRNA transcripts of genes associated with the modulation of the stress response. In the case of acute stressors, a temporal pattern of alterations in *crf*, pro-opiomelanocortin (*pomc*), and glucocorticoid receptor (*gr*) mRNA transcripts has been observed [59,91]. The expression of brain *crf* increases rapidly, approximately 15 min after stress, aligning with a sharp increase in *crf* levels in the preoptic area [59,91]. *Pomc* transcripts reach their peak at 30 min post-stress, indicating a quicker response compared to the previously reported 60 min for pituitary mRNA levels [91]. Additionally, *gr* mRNA transcripts show a peak at 15 min post-stress [59].

The impact of cortisol on molecular, cellular, and physiological levels is mediated by the glucocorticoid receptors (GRs), crucial regulators of the stress response facilitating short- and long-term adaptation and acclimation to adverse stimuli. Disruptions in GR signaling have been associated with impairments in learning and memory processes as well as various brain disorders [93,94,95]. In contrast to several other teleosts where two GR isoforms (GR1 and GR2) have been identified, zebrafish possess a single gene encoding the GR [96,97,98,99,100]. Previous research indicated a swift upregulation (15 min) and a rapid return to baseline levels (30 min) in GR transcripts following exposure of zebrafish to acute stressors [59]. This pattern was consistent in fish exposed to high-intensity, unpredictable chronic stress. These findings align with a recent study in rainbow trout, where short-term cortisol implantation led to an upregulation of GR2 expression in the brain and kidney tissues but a downregulation in other tissues [101].

Brain-derived neurotrophic factor (BDNF), a protein with significance in depression, drug addiction, and aging, also plays an important role in stress resistance and chronic resilience [102]. The expression of the *bdnf* gene in the brain is under the control of GRs [103,104,105,106], and BDNF, in turn, regulates CRH homeostasis [107]. An investigation revealed an increase in *bdnf* transcripts that correlated with the duration of elevated trunk cortisol concentrations in fish subjected to acute stress, reflecting the impact of chronic stress intensity on brain mRNA levels. This suggests the potential use of *bdnf* as an indicator of stimulus specificity and intensity [59]. A similar elevation in brain *bdnf* mRNA levels was observed in adult zebrafish exposed to severe unpredictable stress for 15 days [108].

In conclusion, the exploration of genetic and molecular mechanisms underlying PTSD-like responses in zebrafish has opened avenues for understanding the difficulties of stress-related behaviors. MicroRNAs, cortisol dynamics, and the involvement of key genes such as *crf*, *pomc*, *gr*, and *bdnf* provide valuable insights into the complex networks orchestrating stress responses in this model organism. However, it is crucial to recognize that the field of zebrafish PTSD research is dynamic, with ongoing investigations needed to unravel the full complexity of these mechanisms and their translation to human PTSD.

## 7. Pharmacological and Therapeutic Interventions in Zebrafish PTSD Models

Zebrafish offer a valuable platform for high-throughput pharmacological screening, enabling the identification of compounds that can modulate anxiety-like behavior, fear responses, and other PTSD-like phenotypes [109]. Studies have explored serotonergic drugs, considering the involvement of the serotonergic system in stress responses, and tested drugs targeting serotonin receptors to investigate their effects on anxiety- and stress-related behaviors [110]. Additionally, given the link between neuroinflammation and stress-related disorders, anti-inflammatory agents have been investigated as potential therapeutic interventions, demonstrating beneficial effects on behavior and brain inflammation in stressed zebrafish [111,112]. Anxiolytic compounds, such as benzodiazepines, have also been tested to assess their potential to reduce anxiety-like behaviors in zebrafish [113,114,115]. Furthermore, non-pharmacological interventions like environmental enrichment and enriching living conditions have been explored to mitigate stress-related behaviors in stressed zebrafish [116,117]. It is crucial to emphasize that findings from zebrafish studies must be validated in higher-order animal models and human clinical trials before any potential treatments can be translated to human patients. As the field of pharmacological and therapeutic interventions in zebrafish PTSD models progresses, researchers are encouraged to consult recent research articles and reviews for up-to-date and comprehensive information. The integration of zebrafish models with other preclinical and clinical research holds promise for identifying effective interventions for stress-related disorders, including PTSD.

## 8. Zebrafish in PTSD Research: Validation, Advantages, Limitations, and Future Directions

This review delves into zebrafish stress responses, focusing on fear and anxiety and utilizing established behavioral and physiological metrics. Yet, we stress the importance of delving deeper into chronic or repeated stress models that more accurately mirror the intricate and lasting nature of human traumatic stress. It is worth mentioning existing studies that employ chronic stress paradigms in zebrafish and their potential implications for PTSD research.

Promising models that utilize chronic stress in zebrafish include repeated predator exposure, which has been shown to induce anxiety-like behaviors and gene expression changes relevant to stress response pathways. According to Stewart et al. (2012), this strategy causes anxiety-like behaviors and pertinent gene expression changes, giving valuable insights into stress response pathways [51]; while primarily utilized for depression and anxiety studies, its behavioral alterations may shed light on specific aspects of PTSD related to social withdrawal and isolation, used to explore depression and anxiety but also pertinent to PTSD’s social withdrawal aspects [11,118]. Another promising model is chronic unpredictable stress (CUS), which subjects zebrafish to a variety of unpredictable stressors over time, mimicking human chronic stress conditions closely [38]. This leads to changes in anxiety-related behaviors and neurotransmitter levels, offering a closer approximation of chronic stress experienced in humans [38].

Further investigations could explore combined approaches, integrating chronic stress with factors like social interaction or genetic variations to gain insights into stress vulnerability and resilience. Using a variety of behavioral, physiological, and neurochemical measures in a multidimensional assessment method can give us a more complete picture of how zebrafish react to things that are similar to PTSD. Tailoring stress models to specific traumatic experiences could also enhance the relevance of these models to different PTSD subtypes. While these models show promise, replicating the full breadth of human PTSD in zebrafish is challenging due to the complex interplay of psychological, social, and genetic factors in humans.

Validating zebrafish findings with clinical and other animal studies is essential for understanding zebrafish models’ relevance and potential applications in studying stress-related disorders like PTSD [119]. This validation process involves cross-species comparisons with other animal models, examining similarities and differences in behavioral and molecular responses [120,121]. Translational research connects zebrafish research outcomes to human clinical applications, identifying conserved biomarkers and genetic pathways [122,123]. Comparing zebrafish findings with data from human studies helps establish common genetic, neurobiological, or behavioral aspects [122]. Additionally, in vitro experiments and pharmacological studies further support the validation process [124,125]. By integrating zebrafish findings with data from diverse sources, researchers can gain a comprehensive understanding of the potential clinical implications and utility of zebrafish models in PTSD research. It is important to recognize that while zebrafish models provide valuable insights, a multi-modal approach combining different model systems is essential for robust validation.

The zebrafish serves as a valuable research tool for unraveling the complexities of post-traumatic stress disorders (PTSDs), offering insights into stress and anxiety responses through shared genes and neurobiological pathways with humans [22]. The advantages of utilizing zebrafish in PTSD research are noteworthy. The species exhibits rapid development, allowing for efficient studies with large sample sizes and lower maintenance costs [11,51,126]. Advanced video-tracking tools allow efficient and objective quantification of robust behavioral patterns in zebrafish, enhancing the precision of behavioral observations [109].

Despite these advantages, there are limitations in using zebrafish models for PTSD research. Their cognitive complexity, in comparison to mammals, may not fully replicate the intricacies of human PTSD, given the absence of certain brain structures like the prefrontal cortex associated with higher cognitive functions [109,127]. Additionally, modeling specific human experiences contributing to PTSD poses challenges. Interpreting PTSD-like symptoms in zebrafish requires caution, as fish behavioral responses may not perfectly mirror human emotions and experiences [128].

Nevertheless, zebrafish remains an invaluable complementary model system in PTSD research, providing unique opportunities to study stress-related behaviors and underlying neurobiological mechanisms. The integration of zebrafish findings with data from other model organisms and human studies is crucial for a comprehensive understanding of PTSD complexities and the development of effective therapeutic interventions [125]. By leveraging the strengths of zebrafish research alongside other approaches, scientists can gain comprehensive insights into the pathophysiology of PTSD, potentially identifying novel treatment strategies for this debilitating condition. This positions the zebrafish model as a unique research tool in illuminating the complexities of PTSD. Table 2 provides additional details regarding the advantages and disadvantages of utilizing zebrafish as a model in research.

In zebrafish PTSD research, exploring genetic and molecular mechanisms underlying PTSD-like responses offers opportunities for targeted therapeutic interventions. Advancements in genome-editing technologies, such as CRISPR/Cas9, allow investigation of the specific roles of individual genes in PTSD pathogenesis. Integrating zebrafish research with other model systems and human studies validates findings and enhances translational potential, identifying conserved mechanisms and potential therapeutic targets. Zebrafish models facilitate rapid and cost-effective drug screening for identifying novel compounds for PTSD treatment. Combining behavioral, genetic, and neurobiological investigations is crucial to comprehensively understand the complex nature of PTSD. Leveraging zebrafish models’ strengths and their translational potential contributes to the development of more effective therapies and interventions for individuals with PTSD, extending the model’s applicability to other anxiety disorders and psychological conditions.

## Figures and Tables

**Figure 2 ijms-25-04895-f002:**
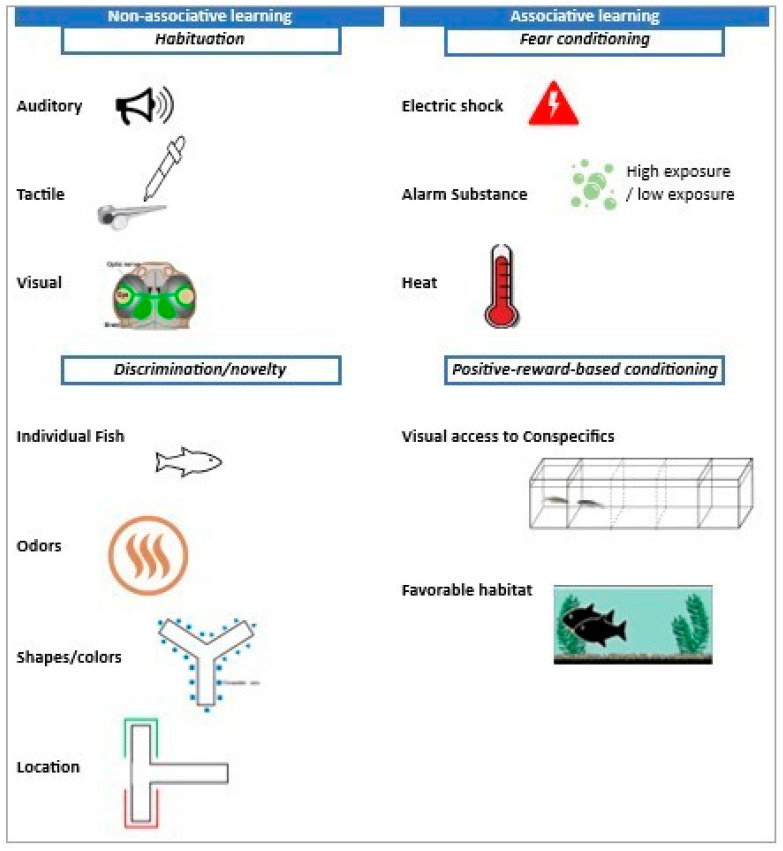
Schematic overview of learning in zebrafish.

**Table 1 ijms-25-04895-t001:** Commonalities between zebrafish and humans in PTSD research.

Key Points	Relevance in PTSD Research
Behavioral Similarities	Zebrafish exhibit fear responses, anxiety-like behaviors, and stress-related responses, making them suitable for studying the impact of traumatic experiences on behavior [23].
Genetic Similarities	Zebrafish share many genetic homologies with humans, allowing researchers to investigate genes associated with PTSD and their impact on the zebrafish model [4].
Neural Circuitry	Zebrafish have a relatively simple and well-characterized nervous system, aiding in understanding neurobiological underpinnings of stress and anxiety responses relevant to PTSD [24].
High-Throughput Screening	The transparency of zebrafish embryos allows for high-throughput drug screening, facilitating the identification of potential compounds or interventions for PTSD treatment [25].
Genetic Manipulation	Various genetic manipulation techniques in zebrafish, such as morpholino knockdowns and CRISPR/Cas9 gene editing, enable researchers to study the roles of specific genes in PTSD-like behaviors and stress responses [11,26,27,28,29].
Environmental Factors	Zebrafish’s adaptability to different stress paradigms allows researchers to induce stress and trauma in controlled laboratory settings to study their effects on behavior and physiology [30].

**Table 2 ijms-25-04895-t002:** Advantages and disadvantages of using zebrafish as a model in research.

Advantages	Disadvantages
Rapid Development Zebrafish exhibit rapid development, enabling the study of PTSD-related changes in a short timeframe.	Behavioral Complexity Zebrafish behavior may not fully capture the complexity of human PTSD symptoms. Thus, zebrafish behavioral responses may not perfectly mirror human emotions and experiences.
Large Sample Size The species allows for studies with substantial sample sizes, enhancing statistical robustness.	Ethical Considerations Large-scale breeding for high sample sizes may raise ethical concerns.
Low Cost Zebrafish maintenance costs are lower compared to larger vertebrate models.	Limited Adult Relevance PTSD typically manifests in mature organisms, limiting the relevance of observations in transparent larvae.
Juvenile Manifestation Symptoms may manifest in the juvenile stage, providing an early window for observation.	Brain Structure Differences Differences in brain complexity and structures may hinder direct translation to human conditions.
Genetic Similarities Significant genetic homology with humans facilitates the exploration of PTSD-related genes.	Limited Predictive Validity Zebrafish models may have limited predictive validity for drug responses in more complex mammalian systems.
Transparent Larvae Transparent larvae enable real-time observation of molecular and cellular changes.	Environmental Sensitivity Susceptibility to stressors may vary, making standardization challenging.

## Data Availability

The data that support the findings in this study are available from the corresponding author upon reasonable request.

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
