# Peer review of "Zebrafish Model in Illuminating the Complexities of Post-Traumatic Stress Disorders: A Unique Research Tool"

_ijms, 2024, doi:10.3390/ijms25094895_

Round 1
Reviewer 1 Report (New Reviewer)
Comments and Suggestions for Authors
PTSD is a psychological condition that typically develops in response to traumatic events in humans, and it is challenging to replicate this condition in animals. Creating a PTSD model in mice, for instance, is not straightforward.
This review article discusses specific aspects of stress, fear, and anxiety in zebrafish, along with relative behavioral analyses, aiming to provide insights into the neural and physiological mechanisms underlying PTSD. However, due to zebrafish having less developed cognitive and emotional capabilities compared to humans, it poses a challenge to directly relate stress in zebrafish to the induction of traumatic stress.
The central question here is how to create an authentic PTSD model in zebrafish. It would be beneficial if the author could explore experiments involving chronic or prolonged stresses in zebrafish, which might better mimic the conditions associated with PTSD. Perhaps the author could discuss studies that have attempted such approaches in zebrafish.
According to PubMed, there are only 14 publications using the keywords "zebrafish" and "PTSD." Many of these publications have not been cited or discussed in the current manuscript. It might be worth the author's consideration to incorporate these studies into their discussion.
One minor correction: On line 65, "Post-Traumatic Stress Disorder," the first letter should not be capitalized.
Author Response
Thank you for your valuable comments. Please find attached the document containing our response to each of the provided comments.

Reviewer 2 Report (New Reviewer)
Comments and Suggestions for Authors
The review presented by Al-Zoubi et al., can be a valuable source of information on modeling PTSD in Zebrafish Model. The authors consider the possibility of such modeling from the point of view of various aspects: behavioral, pathophysiological, molecular genetic, pharmacological. The manuscript is well structured and contains all the relevant information. However, I have a few comments.
1. In my opinion, Figure 2 (a schematic overview of zebrafish training) does not fully reflect the content of Section 6 (Post-Traumatic Stress Disorder (PTSD) in Zebrafish: Behavioral Aspects). Thus, its lower half is not explained in any way in the text. This figure should either be removed or further explanation given.2. It is incorrect to call BDNF a peptide; it is a protein.
Author Response
Thank you for your valuable comments. Please find attached the document containing our response to each of the provided comments.

Reviewer 3 Report (New Reviewer)
Comments and Suggestions for Authors
In this comprehensive review from Al-Zoubi et al., the authors describe the use of zebrafish as a potential and innovative tool in PTSD research, highlighting pros and cons of this animal model. Minor suggestions are reported here.
1. For clarity, I would suggest rephrasing the sentence starting in line 43 (i.e., Individuals who fail to recover from PTSD can experience ...)
2. Line 52 - I would change it to "easier to quantify," or if you prefer "easily," it should be "more easily quantifiable."
3. Space is missing in line 89 where reference 20 is cited.
4. Line 93 - I would add a full list of the model's limitations (i.e. anatomical differences in the CNS, limited availability of research tool such as antibodies for testing, presence of duplicate genes that could make it hard to be modeled/silenced).
5. The addition of a figure that shows zebrafish - i.e. during its development cycle - could add more visual impact to the paper.
6. Line 113 & line 147- HPA/HPI acronyms should come before the word axis.
7. Line 115 - The sentence could be developed more by adding a short description of what cortisol is. An idea could be to move the description from lines 121-124 here and merge them into just one sentence.
8. Lines 124-125. This sentence is not clear. Please consider rephrasing it.
9. Line 147 - sentence could use rephrasing (i.e., the HPI axis in zebrafish has a similar role in stress response as the HPA axis in humans).
10. Line 151 - please add "to" after "cortisol,".
11. Line 172 - It is not clear if we are still talking about zebrafish neural circuits or humans. Please clarify.
12. Line 175 - No need to repeat the acronym HPA in full length.
13. Line 184 - It would make it easier to read if the final "comprehensively" would be removed and replaced with "in needed to better elucidate".
14. Line 239 - To avoid pronouns issues, it would be better to replace "Blaster and his colleagues" with "In this previous study, it was found...".
15. Line 243 - Add a space after pre-, please.
16. Line 252 - If authors are referring just to chemical exposures, please consider specifying that here.
17. Lines 255-256 - The acronyms for the NTT and LDT tests have already been explained in line 212. Please report them in full length only once, and please choose between NTT or NTDT/ LDT or LDPT, considering it seems they are referring to the same tests.
18. Line 263 - Please consider adding "previous" (i.e. In a previous study).
19. Lines 307-311 - This could be merged in one sentence, by replacing zebrafish (line 309) with "as they".
20. Line 364 - Do the authors mean HPI?
21. Line 377 - there is a typo. Please change "sumarises" with "summarizes".
22. Paragraph 7 has a different formatting than the rest of the paper. Please uniform this.
23. Line 384 - for clarity, I would recommend adding "several studies".
24. Line 400 - Please add a space after reference 78.
25. Line 401 - the use of "individuals" when referring to zebrafish can be confusing, considering the comparison to human happening in other parts of the paper. Please consider changing it.
26. Line 412 - Please add here the full name for CRF and remove it from line 415.
27. Line 437 - Please add the full name for CRH.
28. Line 487 - For clarity, I would recommend adding " with humans after "zebrafish share..."
Comments on the Quality of English Language
English could be improved in some sentences. Please refer to the "Suggestions" section.
Author Response
Thank you for your valuable comments. Please find attached the document containing our response to each of the provided comments.

Reviewer 4 Report (New Reviewer)
Comments and Suggestions for Authors
In the review article “Zebrafish Model in Illuminating the Complexities of Post-2 Traumatic Stress Disorders (PTSD): A Unique Research Tool,” the authors try to justify that the zebrafish may be an adequate model for an in-depth study of post-traumatic stress disorder and provide an analysis of a large literature, in which they are trying to prove this idea. While citing a large number of arguments to prove the effective use of the fish model for assessing behavioral reactions in humans, there is nevertheless no need to oversimplify the comparability of this model using the example of fish and reactions in humans. This approach overly simplifies the view of the study of this issue and requires more careful formulations, so I advise the authors to use more careful formulations in the abstract and introduction, not forgetting that the behavior of fish is largely instinctive, and to draw more careful analogies when comparing behavioral reactions of fish and humans.
1. The Summary section needs to be revised in accordance with the comments made.
2. The Reference No. 1 is not clear; it is necessary to provide a complete version of a correctly formatted reference.
3. Authors must provide full versions of the published articles to which they refer, and not indicate only the 1st author. Errors occur in 1, 5, 6, 8, 9, 10, etc. Thus, the entire list of references must be redone and formatted according to the rules of the journal.
4. The review article must contain at least 50% of references from the last 10 years.
5. In the Introduction section, the authors conclude “that the zebrafish genome has been fully sequenced and is 87% similar to the human genome. Approximately 70% of the 56 zebrafish genomes are orthologous to the human genome. These similarities mean that zebrafish also harbor numerous genes associated with human diseases, allowing mutations observed in humans to be studied using the zebrafish model.” The analogy between the human genome and the zebrafish genome needs to be drawn more carefully, and the authors should not oversimplify these analogies or make more gentle assumptions based on evidence-based experimental data and caution in suggesting that the data will be used to “improve our understanding of post-traumatic stress disorder.” Because the authors themselves write that zebrafish do not suffer from post-traumatic stress disorder in the same way as humans.
6. The authors need to be critical when comparing the manifestation of stress reactions in zebrafish and in humans. It might be useful to create a comparison table summarizing the advantages and disadvantages of using zebrafish as a model of PTSD.
7. In the section “Pathophysiology of PTSD,” the authors should more clearly analyze the pathophysiology of stress disorder in zebrafish and humans. In this edition, it is not clear what exactly the authors want to emphasize when providing details of short-term stimulation of the human hypothalamic-pituitary-adrenal axis (refs. 30-37) and why they provide information about cavefish when characterizing hormonal effects on zebrafish. It is also not clear why these data are limited to only one species and ignore data on other species, such as salmon?
8. The authors need to clarify for which species Scheme 1 is presented. It would be nice to present data not only on humans, but also in a comparative aspect for zebrafish.
9. Sections 4, “Neuroanatomical and neurochemical evidence from studies of PTSD in zebrafish,” and 5, “Behavioral paradigms for assessing PTSD in zebrafish,” are very fragmented and inconsistent with the progress of the field of research.
10. The final sections, including “Validation of Zebrafish Findings through Clinical and Other Animal Studies,” “Strengths and Limitations of Zebrafish in PTSD Research,” and “Future Directions and Translational Potential in Zebrafish PTSD Research,” are recommended to be combined into one final section, in which all the listed fragments will be presented, since individually they are too fragmentary, as independent sections and have an internal logical connection with each other.
Comments on the Quality of English Language
English language needs correction
Author Response
Thank you for your valuable comments. Please find attached the document containing our response to each of the provided comments.

Round 2
Reviewer 4 Report (New Reviewer)
Comments and Suggestions for Authors
Authors made significant corrections to the original version of the work, but did not correct the comments on points 2, 3 in the review. Also, the comments on point 8 are not very convincing. It is recommended to make final corrections to the work in accordance with the recommendations of the review.
Comments on the Quality of English LanguageEnglish language correction is recommended
Author Response
Thank you for your valuable comments and suggestions. Please find attached our response to your comments

This manuscript is a resubmission of an earlier submission. The following is a list of the peer review reports and author responses from that submission.
Round 1
Reviewer 1 Report
Comments and Suggestions for Authors
The authors reviewed zebrafish models for PTSD, in terms of compatibilities related to neuronal function, fear response and molecular mechanisms. Topics related to zebrafish models to study brain function and neuropsychiatric disorders seem scientifically attractive. Sever stress is one of the main triggers of PTSD and various kinds of stresses may trigger the PTSD. So, the authors should carefully summarize what kinds of stresses cause behavioral and molecular alteration in zebrafish. However, the authors describe effects of stress that seem unrelated to PTSD and it is not clear that which zebrafish models are compatible and scientifically reliable models to study PTSD. Moreover, more than ten inappropriate citations (e.g., irrelevant citations, duplicate citations), were found especially in the molecular mechanisms section.
Comments on the Quality of English LanguageThe quality of English language is almost enough quality. Some extra space, and ambiguous use of uppercase and lowercase letters regards to zebrafish were found.
Author Response
Dear Respected Editors
We greatly appreciate the reviewers' comments on Manuscript ID: ijms-2606886, title: Zebrafish Model in Illuminating the Complexities of Post-Traumatic Stress Disorders (PTSD): A Unique Research Tool. We have carefully addressed the comments and revised the manuscript accordingly.
Our responses to the reviewers are given in a point-by-point manner (blue color), and changes to the manuscript are uploaded as a 'revised Manuscript' file; a clean version of the main Article File is provided. We hope that you find the paper now acceptable, below is the reply to the reviewer's specific Comments:
Reviewer 1
The authors reviewed zebrafish models for PTSD, in terms of compatibilities related to neuronal function, fear response and molecular mechanisms. Topics related to zebrafish models to study brain function and neuropsychiatric disorders seem scientifically attractive.
- Sever stress is one of the main triggers of PTSD and various kinds of stresses may trigger the PTSD. So, the authors should carefully summarize what kinds of stresses cause behavioral and molecular alteration in zebrafish.
Thank you for the valuable comment, the below paragraph was added to the manuscript to summarize kinds of stresses cause behavioral and molecular alteration in zebrafish.
Zebrafish can exhibit behavioral and molecular alterations in response to various stressors [1]. Physical stressors like temperature fluctuations and water quality changes can impact swimming patterns and gene expression [2]. Chemical stressors, including pollutants and toxins, disrupt physiological processes and trigger molecular stress responses [3]. Social stressors, such as aggression and social interactions, affect behavior and neural pathways related to social behaviors [4]. Psychological stressors like isolation and novel environments activate the stress response system, influencing behavior and gene expression [5]. Infections and pathogens induce stress and alter immune responses, leading to molecular changes [6]. Nutritional stress affects growth, reproduction, and metabolism, resulting in behavioral and molecular adaptations [1, 2]. Long-term exposure to environmental changes can lead to chronic stress, causing persistent behavioral and molecular adjustments [7, 8]. These responses are often adaptive and aimed at helping the fish cope with the specific stressor and maintain homeostasis [1]. Researchers often study these responses in zebrafish to gain insights into stress biology and potential human health implications.
- However, the authors describe effects of stress that seem unrelated to PTSD, and it is not clear that which zebrafish models are compatible and scientifically reliable models to study PTSD.
Thank you for the valuable comment, the below paragraph was added to the manuscript to clarify that zebrafish models are compatible and scientifically reliable models to study PTSD.
Zebrafish have become valuable models in the study of stress-related disorders like Post-Traumatic Stress Disorder (PTSD) due to their genetic tractability, rapid reproduction, and shared physiological and molecular pathways with humans [9]. While zebrafish do not experience PTSD in the same way humans do, they exhibit stress responses that make them suitable for studying the underlying mechanisms and potential treatments [10]. Zebrafish serve as scientifically reliable models for studying stress-related disorders like PTSD due to their genetic similarities to humans, shared physiological pathways, and observable stress responses [1, 11]. When exposed to stressors, zebrafish exhibit hormonal changes, altered behaviors, and neural circuitry responses that mimic aspects of human stress responses [2]. Their transparent embryos and larvae are particularly useful for studying early-life stress and neural development [12]. Additionally, zebrafish models offer ethical advantages, as they allow for genetic and pharmacological manipulations with fewer ethical concerns than mammalian models[13]. While zebrafish may not fully replicate the cognitive and emotional aspects of PTSD, they provide valuable insights into the underlying mechanisms and potential treatments, complementing research conducted in other species and humans [14].
Despite these advantages, it's important to note that zebrafish models have limitations. They lack the complex cognitive and emotional aspects of PTSD that humans experience [14]. However, they provide a valuable platform for studying the molecular and physiological underpinnings of stress responses and can complement research conducted in other animal models and humans.
In summary, zebrafish models are scientifically reliable tools for investigating the physiological and genetic aspects of stress responses related to PTSD. While they may not fully replicate the human condition, their advantages make them valuable contributors to our understanding of the mechanisms underlying stress-related disorders and the development of potential treatments.
- Moreover, more than ten inappropriate citations (e.g., irrelevant citations, duplicate citations), were found especially in the molecular mechanisms section.
All references were revised and fixed as recommended.
Comments on the Quality of English Language
- The quality of English language is almost enough quality. Some extra space, and ambiguous use of uppercase and lowercase letters regards to zebrafish were found.
Uppercase and lowercase letters regard to zebrafish were revised and fixed as recommended.

Reviewer 2 Report
Comments and Suggestions for Authors
This review provides a full picture of using zebrafish as a model to study PTSD. The topic is interesting and the manuscript will be helpful to the field.
The major problem is that many references are not the original research or the most relevant ones. For example, line 52-53 zebrafish-human genome comparison, authors may want to cite [10] instead of [5] and [6]. The other example, in table1 when introducing genetic manipulation in zebrafish, one would imagine citations about the first application of morpholino or CRISPR in zebrafish, or the best-summarized method paper, or the zebrafish PTSD research using such methods. Given there might not be enough zebrafish PTSD research out there, zebrafish behavioral studies would be more relevant than zebrafish embryo development (ref 13, morpholino or CROSPR not used in this paper), Cardiotoxicity (ref 14) or heart development (ref 15).
Minor problems:
Font size is bigger in the second paragraph of abstract.
Many unnecessary “zebrafish” in upper case throughout the text.
Some references are not in correct format, for example ref 1, 26, 115, 149.
Line 56, two periods.
Comments on the Quality of English LanguageEnglish is good.
Author Response
Dear Respected Editors
We greatly appreciate the reviewers' comments on Manuscript ID: ijms-2606886, title: Zebrafish Model in Illuminating the Complexities of Post-Traumatic Stress Disorders (PTSD): A Unique Research Tool. We have carefully addressed the comments and revised the manuscript accordingly.
Our responses to the reviewers are given in a point-by-point manner (blue color), and changes to the manuscript are uploaded as a 'revised Manuscript' file; a clean version of the main Article File is provided. We hope that you find the paper now acceptable, below is the reply to the reviewer's specific Comments:
Reviewer 2
This review provides a full picture of using zebrafish as a model to study PTSD. The topic is interesting, and the manuscript will be helpful to the field.
- The major problem is that many references are not the original research or the most relevant ones. For example, line 52-53 zebrafish-human genome comparison, authors may want to cite [10] instead of [5] and [6]. The other example, in table1 when introducing genetic manipulation in zebrafish, one would imagine citations about the first application of morpholino or CRISPR in zebrafish, or the best-summarized method paper, or the zebrafish PTSD research using such methods.
All references were revised and fixed as recommended.
- Given there might not be enough zebrafish PTSD research out there, zebrafish behavioral studies would be more relevant than zebrafish embryo development (ref 13, morpholino or CROSPR not used in this paper), Cardiotoxicity (ref 14) or heart development (ref 15).
Unrelated references were removed and replaced with more relevant ones.
Minor problems:
- Font size is bigger in the second paragraph of abstract.
The font size was unified through the text.
- Many unnecessary “zebrafish” in upper case throughout the text.
Done, all unnecessary “zebrafish” uppercase throughout the text was addressed.
- Some references are not in correct format, for example.
- Ref 1, This reference dissertation for a degree Master of Science in Pharmacology at the Northwest University.
- Ref 26, This reference is a book, and the format was corrected.
- Ref 115, This reference was duplicate with reference 109 and the format was corrected.
- Ref The format was corrected.
Line 56, two periods.
Done, one period was removed.
Round 2
Reviewer 1 Report
Comments and Suggestions for Authors
Previous comments shown bellow have not been enough corrected, because the authors just revised the citations.
"Moreover, more than ten inappropriate citations (e.g., irrelevant citations, duplicate citations), were found especially in the molecular mechanisms section."
Authors mentioned Cd11b, IL4 and other molecules related to PTSD in zebrafish, but no citations showing that PTSD-related stress induced the mentioned molecules in zebrafish.
Previous draft indicated no COI, but the revised draft has been changed as shown bellow.
"Conflicts of Interest: The authors declare that this study received funding from Z (please replace Z by the funder name). The funder was not involved in the study design, collection, analysis, interpretation of data, the writing of this article or the decision to submit it for publication. "
